# Nutrition Education Programs Aimed at African Mothers of Infant Children: A Systematic Review

**DOI:** 10.3390/ijerph18147709

**Published:** 2021-07-20

**Authors:** Cristina Jardí, Byron David Casanova, Victoria Arija

**Affiliations:** 1Nutrition and Public Health Unit, Research Group on Nutrition and Mental Health (NUTRISAM), Faculty of Medicine and Health Science, Rovira i Virgili University, 43201 Reus, Spain; cristina.jardi@urv.cat (C.J.); byrondavid.casanova@estudiants.urv.cat (B.D.C.); 2Pere Virgili Institute for Health Research (IISPV), Rovira i Virgili University, 43003 Tarragona, Spain

**Keywords:** malnutrition, stunting, wasting underweight, nutrition programs, systematic review

## Abstract

Background: Child malnutrition is a major epidemiological problem in developing countries, especially in African countries. Nutrition education for mothers can alleviate this malnutrition in their young children. The objective of this study was to make a systematic review to assess the effect of intervention programs in nutrition education for African mothers on the nutritional status of their infants. Methods: A bibliographic search was carried out in the PubMed database for clinical trials between November 2012 and 2021. The studies should contain educational programs to evaluate the impact on the infant’s nutritional indicators in children under 5 years (food consumption, anthropometry and/or knowledge of nutrition in caretakers). Results: A total of 20 articles were selected, of which 53% evaluated infant’s food consumption, 82% anthropometric measurements and 30% nutritional knowledge. In general, nutritional education programs are accredited with some significant improvements in food and nutrient consumption, knowledge and dietary practices in complementary feeding, but only those studies that implemented strategies in agriculture, educational workshops and supplementation obtained reductions in chronic malnutrition figures. Limitations: There is high heterogeneity in the articles included, since the intervention programs have different approaches. Conclusions: Programs that implemented actions of national agriculture or nutritional supplementation reap the greatest benefits in curbing infant malnutrition.

## 1. Introduction

Child malnutrition is a worrying public health problem in developing countries. According to 2019 statistics, it is estimated that worldwide, one in five children under 5 years of age has some degree of malnutrition, about 150 million suffer from insufficient stature and almost 50 million are wasted [1]. The most disadvantaged continents are Africa, Asia and Oceania [1,2]. Of these, Africa has shown alarming increases in the rates of chronic malnutrition: the 22.4 million cases in 2000 rose to 28.9 million in 2018 [2]. To the main underlying drivers of food insecurity, such as climate change, conflict and economic recessions, the impact of COVID-19 has recently been added, disproportionately affecting the African continent [3]. Children under the age of five are the most affected by malnutrition in Sub-Saharan Africa [4]. The burden of malnutrition has been directly linked to poverty, quality of food intake, excessive disease and poor health status [5]. Pregnant women, lactating women and children of infant age are the most nutritionally vulnerable groups. They have considerable nutritional requirements because they are growing, but these requirements are difficult to meet, which leads to poor weight gain in children and during pregnancy. In turn this leads to low birth weights of their babies, which is aggravated by multiple pregnancies in short periods of time and by the high frequency of HIV infections. Therefore, both mothers and children on this continent are affected, and they also have to cope with a low educational level [6].

On the other hand, it has been observed that good health care practices, better nutrition and living conditions and greater general knowledge of the mother have a positive effect on maternal and child health [7], while the precarious availability of food and a low educational level of the mother are factors that increase the risk of infant morbidity and mortality from preventable causes [8].

Various studies have suggested that providing mothers with educational strategies about feeding has a significant positive impact on their children’s nutritional status, especially when they are integrated into the local contexts they are designed for. Dewey and Adu-Afarwuah in a systematic review (2008) [9] evaluated the different types of educational interventions in various developing countries (educational interventions, provision of food offering extra energy (with or without micronutrient fortification), micronutrient fortification of complementary foods, increasing energy density of complementary foods through simple technology) in the period from 1996 to 2006 and found generally positive effects in Asia and Africa, although the results depended on the availability of food in the environment. In a subsequent systematic review, Imdad et al. (2011) [10] evaluated the growth and weight of young children in terms of whether the feeding programs had been carried out with or without instruction in complementary feeding on children less than 2 years of age for mothers from developing countries. They observed that although the results in weight and height gain were similar in both groups, there was a need to obtain more evidence before any conclusions could be drawn in this regard.

The present study aims to carry out a systematic review to assess the effect of intervention programs in nutrition education, designed for African mothers, on the nutritional status of children of infant age.

## 2. Materials and Methods

### 2.1. Search Strategy

Following PRISM Systematic Review Guidelines, a bibliographic search was carried out in the PubMed database of clinical trials published between November 2012 and 2021, the last date of search being 15 June 2021, using the MESH terms with the following combination:(((programs OR intervention) AND education AND (nutrition OR diet OR food OR feeding OR malnourished OR wasting OR “Wasting Syndrome”[Mesh] OR malnutrition OR “Protein-Energy Malnutrition”[Mesh])) AND ((mother OR maternal OR women) AND (child*))) AND (Africa OR “Africa South of the Sahara”[Mesh] OR “South Africa”[Mesh] OR “Mozambique”[Mesh]).

### 2.2. Selection Criteria

Clinical trials were included that carried out intervention programs in nutrition education for mothers or caregivers of children less than 5 years of age who were African and malnourished. Those focused on educational programs were to be selected, as well as those that, in addition to being educational, are complemented with the promotion of domestic agriculture, food/nutrient supplementation or similar characteristics.

Moreover, the programs had to be evaluated by indicators that assessed the knowledge and skills acquired by mothers or caregivers about infant feeding, infants’ anthropometric development and nutritional status (food consumption and nutritional intake).

We did not apply any language restriction.

Manuscripts that did not have the above characteristics were excluded, as were those focused on children or mothers with specific diseases (e.g., HIV), non-health education-based programs, non-Africans and children over 5 years of age.

### 2.3. Selection of Articles

All data from the eligible clinical trials were independently abstracted by two researchers (BC and VA) without significant disagreements. The screening of titles and abstracts for all studies was independently performed by two authors (BC and VA) in order to identify potentially relevant articles. Likewise, these two authors (BC and VA) performed full-text screenings and with mutual consensus, confirmed that studies met the study’s inclusion and exclusion criteria.

Subsequently, the bibliographic citations referenced in the articles selected were reviewed to locate other investigations (CJ). The authors resolved discrepancies after discussion articles were not excluded for reasons of language.

The quality of the studies selected was examined by the ranking of the journals published through the web InCites Journal Citation Reports and referenced by quartiles (Q) in *Nutrition & Dietetics*.

### 2.4. Data Abstraction

In data management, each study was taken and evaluated by the year of publication and the country where it was carried out, the educational guides used and type of intervention (supplementation, agriculture, education), sample size and duration of the intervention, evaluated indicators and quantification of the results. Every study should evaluate baseline vs. final line; we took the results with a significance value lower than 0.05.

## 3. Results

A total of 122 articles were retrieved from the PudMed search. From these, 41 were selected on the strength of the title, of which 10 were excluded due to the content of the abstract and 11 after a complete reading of the text. Subsequently, five studies were included after a manual search to complete a selection of 20 articles (see Figure 1).

Table 1 describes the characteristics of the studies selected and gives information about the type of intervention and their methodology, nutritional assessment indicators used, results obtained and the quality quartile of the article.

Virtually all of the selected studies are randomized clinical trials, except for three [37,45,52], which are quasi-experimental. There are three types of programs: those that use only pedagogical models and kitchen practices (EP) [11,13,14,17,22,24,35,45,47,50,51], those that use pedagogical methods and encourage domestic agriculture (AEP) [20,32,39] and those that use pedagogical methods and food supplementation (ESP) [27,31,37,43,46]. Validated educational models were mainly provided by the World Health Organization (WHO) [24,35,50,51,52], United Nations Children’s Fund (UNICEF) [14,32], Food and Agriculture Organization (FAO) [14,32], Positive Deviance/Hearth [27,31,37], the Health Belief Model [17,22,47] and Alive and Thrive [11,22]. Other programs used guides or manuals designed by the researchers [39,45,52].

Some of the nutrition education programs were implemented by mothers who are leaders in the environment [20,45] and others by health teams or health professionals [13,14,17,20,22,27,31,35,37,47,50,51], community volunteers [17,27,31,32,43] or a mixture of both [17,20,27,31].

The effect of the program on mothers or infants was assessed by various indicators. Several EPs have used different questionnaires to evaluate eating activities and the knowledge acquired by the mothers in food hygiene, food selection and complementary feeding [11,13,14,17,22,24,45,47]. They have also used indicators of infant feeding such as minimum dietary diversity (MDD), which assesses the consumption of four or more food groups, and minimum meal frequency (MMF), which assesses the proportion of breastfed children aged 6 to 23 months of age who received food a minimum number of times or more per day. Other programs (AEP, ESP) used these indicators: MDD and MMF [31,32] or knowledge of food nutrition [20]. Only four studies measured intake using dietary records [11,31,39,45]. Anthropometric development was the most used assessment in EP [11,14,22,24,35,45,50,51], AEP [20,32,39] and ESP [27,37,43,54]. It was assessed by such indicators as changes in weight or arm circumference and those provided by the WHO: weight/height or length (W/H or W/L), height/age (H/A) and weight/age (W/A). These indicators were used to estimate the degree of malnutrition: acute malnutrition (<−2 DS z-score W/H or W/L), chronic malnutrition (<−2 DS z-score H/A) and underweight (<−2 DS z-score W/A).

## 4. Discussion

Although there are a considerable number of nutritional interventions in Africa to alleviate infant malnutrition, this review has found few clinical trial-based nutrition education intervention programs. Although our analysis has shown some beneficial effect in educational programs on the nutritional status of children, it is not enough to generate nutritional changes in children. In order to observe which programs have had better results, the discussion was structured on the effect observed on the analysed result variables, first, over knowledge and feeding practices acquired by mothers, together with the nutritional intake carried out by the infant, and second, on the infant’s anthropometric development and degree of malnutrition.

### 4.1. Effects of Educational Programs on Mother’s Knowledge and Feeding Practices, and Infant Nutritional Intake

The main objective of advanced studies on nutrition education is to bring about a change in the attitudes and beliefs of African mothers about infant feeding and to increase their understanding [11,14,46]. Dietary practices and knowledge of nutrition and infant feeding were evaluated differently in each study: some of the tools were developed or adapted by the researchers, and they used various models and methodologies. Even so, despite these differences, the understanding and practice of complementary feeding underwent significant positive changes in the intervention groups of the EPs [13,14,22,45]. The ESP and AEP did not assess nutritional knowledge except for the program directed by Olney et al. (2015) [20], the result of which was clearly favourable. Therefore, nutritional educational programs can reinforce the understanding that mothers and/or caregivers have of child growth and development. Leroy et al. (2020) [54] observed anthropometric improvement using a three-treatment arm randomized clinical trial with different nutritional inputs. A clinical study and a meta-analysis [55,56] corroborate the results obtained and indicate that nutritional education promotes appropriate practices in infant feeding and improves the nutritional status of children.

In this review, we try to analyse the feeding practices and nutritional knowledge of the different types of programs. We found that the food and nutritional intake of infants in the EP was evaluated by analysing nutritional intake and using the MMF and MDD. However, only two clinical trials conducted in Ethiopia and Kenya analysed nutritional intake [11,45]. In the Kenya study [45], after a short period of 5 days, no effect on protein, calorie or iron intake was observed, although there was a significant increase in the intake of vitamin A and calcium [45]. In Ethiopia, the preparation of children’s recipes with beans and grains led to a significant increase in the intake of vegetable protein (7 g/day) and iron (10 mg/day) in children between 6 and 23 months old [11]. Another study in non-African countries—a randomized clinical trial in Peru—observed favourable results in the consumption of micro and macro nutrients after an educational program on complementary feeding from birth to 18 months of age for mothers of children younger than 24 months. Intakes of energy, protein, iron and zinc were observed to be higher in the intervention group [57]. The results were good despite the fact that the program did not include activities on regional agriculture, which suggests that not facilitating access to food did not harm the positive effects because access was not a regional limitation. Some of the studies reviewed here reinforce this hypothesis [14,17,31] and show that the WHO Infant and Child Feeding Indicators (MDD and MMF) are favourably related to the implementation of EPs in Africa. However, although these infant feeding indicators do not report nutrient intake, they serve as approximations of compliance with energy and micronutrient requirements in children who are vulnerable to malnutrition.

Of the programs that encouraged domestic agriculture, only one study analysed nutrient intake and found an association with the increase in nutritional intake in children. The intervention was carried out on farm families with children between 36 and 72 months of age who were involved in an educational program that included agricultural practices. The intervention consisted of a daily dose of multiple-micronutrient powder for 3 months. An improvement was observed in the intake of calories (1627 kcal vs. 1376 kcal, *p* ≤ 0.001), proteins (54 g vs. 48 g, *p* ≤ 0.05) and iron (13 mg vs. 12 mg, *p* ≤ 0.05) in children from the intervention groups [43]. Very few studies evaluated food intake in programs that encouraged domestic agriculture, in addition to the nutrient intake analyses, and only one study reports an increase in MDD (61.9 to 71.1% vs. 59.9 to 55.5%; *p* ≤ 0.01) [32]. Similar results were found in a systematic review in Vietnam, which examined the effectiveness of programs designed to improve dietary intake, household economy and food production in farm families to reduce malnutrition in children under 5 years of age. The results for these variables were significantly positive, and the consumption of foods of animal origin in the general Vietnamese population and the production of sources of protein of animal origin have increased in the last 10 years [58]. Despite not observing clear evidence, in the results of previous studies conducted in Africa, experts recommend the importance of reinforcing the nutritional education of mothers, encouraging the implementation of monitoring programs for child growth, postnatal visits, domestic agriculture and other adjustments to dietary practices so as to increase dietary diversity and the frequency of meals [58,59,60].

No studies have reported on nutrient intake in the programs that use micronutrient supplementation and food fortification (ESP). Despite not reporting intake, the 3-month program carried out in Mali by Somassè et al. (2018) [43], based on nutritional education and micronutrient supplementation, observed a significant increase in haemoglobin concentration and decrease in the prevalence of severe anaemia in the intervention group, unlike the control group. The control group received training while the intervention group received training along with a pack of micronutrients per day consisting of 400 µg vitamin A (retinol equivalents), 150 µg folic acid, 5 µg cholecalciferol (vitamin D3), 90 µg iodine, 17 µg Se, 0.9 µg vitamin B12, 6 mg niacin, 10 mg Fe, 4.1 mg Zn, 0.56 mg Cu, 0.5 mg thiamine, 0.5 mg riboflavin, 30 mg vitamin C, 0.5 mg vitamin B6 and 5 mg vitamin E. The nutritional intake of the intervention group was close to the WHO recommendations in Fe, vitamin A and Zn per package.

In summary, all the studies that implemented educational workshops, strategies in agriculture and supplementation obtained similar results on nutritional and food intake, mainly in terms of calories, protein, iron, calcium and vitamin A [11,14,17,32,39,45]. This is important considering that these nutrients are key in the nutritional problems in developing countries [55]. Similarly, intake indicators such as MDD and MMF increased favourably. This suggests that nutrition education accompanied by other strategies increases nutritional intake or prevents nutrient deficiency. However, it should be noted that, for effective comparison, the parameters of intake measurements in the studies should be standardised.

### 4.2. Effects of Educational Programs on Anthropometric Development and Degree of Malnutrition

The programs that exclusively taught educational workshops obtained few results on anthropometric development. Muhoozi et al. (2019) [35] published three papers that evaluated anthropometric effects after providing education, information and preparation techniques for complementary feeding on the basis of WHO guidelines. Anthropometric results were only significant for H/A indicators in participants up to 36 months of age (intervention z-score −2.15 vs. control −2.65; *p* ≤ 0.01). Despite this, the percentages of chronic malnutrition did not decrease significantly: in the intervention group, they decreased slightly from 20.9 to 18.1%, while in the control group, they increased from 28% to 36% [51]. However, two programs based on the educational strategy of the Health Believe Model, the Alive-Thrive manual and Positive Deviance (PD)/Hearth obtained favourable results in other short-term nutritional indicators such as W/A and W/H [22,27]. The Alive-Thrive strategy is a global initiative to combat malnutrition, save lives, prevent disease and ensure growth and development through optimal maternal nutrition, breastfeeding and complementary feeding practices. It is based on evidence frameworks, social support networks, personalized counselling, massive campaigns and community mobilization [61]. One of the four studies on nutrition education workshops reviewed here, which used messages and materials from the Alive Thrive initiative in Ethiopia and the underlying strategy of the Health Believe Model [22], managed to achieve primary objectives, such as improving infant feeding practices through changes in understanding and attitudes, and secondary objectives, such as determining changes in the anthropometric development of young children aged 6 to 18 months after a 6-month intervention (endline intervention vs. control W/H 2, 20 * vs. 0.89 * and W/A 0.31 * vs. −0.73 * *p* ≤ 0.05). The Health Believe Model is known to be a useful tool in social psychology for predicting and generating health-related behavioural changes. It is based on the perception of health benefits through results that demonstrate the safety of the practices that will lead to those results. The constructs are perception of susceptibility, perception of severity, perception of benefits, perception of barriers, modification of variables, action signals and self-efficacy [62]. Kang et al. (2017) [27] used the Positive Deviance (PD)/Hearth in an educational program that involved supplementation with food and local recipes such as mixtures of soybeans and corn. Compared with children 6 to 24 months of age in the control area, those in the intervention area had a greater increase in z scores for length-for-age [difference (diff): 0.021 z score/month, 95% CI: 0.008, 0.034] and weight-for-length (diff: 0.042 z score/month, 95% CI: 0.024, 0.059). At the end of the 12-month follow-up, children in the intervention area showed an 8.1% (*p* = 0.02) and 6.3% (*p* = 0.046) lower prevalence of stunting and underweight, respectively, after controlling for differences in the prevalence at enrolment, compared with the control group. The programme was effective in improving child growth and reducing undernutrition in this setting [27] The Positive Deviance/Hearth, which is a guide for implementing programs for the prevention and reduction of child malnutrition, was based on a rigorous process of inquiry into the problems and potentialities within the community, thereby using local wisdom to successfully treat and prevent malnutrition. In the Hearth approach, which must be locally adapted, community volunteers encourage the caregivers of malnourished children to learn and practice new calorie-dense recipes, hygiene care, feeding skills and nutritional rehabilitation through group activities followed by home visits. A systematic review analysed the worldwide effects of the Positive Deviance/Hearth program, and 15 out of 17 studies showed positive nutritional and/or behavioural effects particularly when adjusted to the setting of each study, so it has proved to be a useful tool not only for rehabilitation but also for the prevention of malnutrition globally [63]. A doctoral thesis carried out in the district of Ibo in the province of Cabo Delgado (Mozambique) shows that an intensive educational program for mothers with children suffering from moderate acute malnutrition and the implementation of nutritional supplements based on local foods managed to significantly reduce chronic malnutrition after 3 months of follow-up (from 48.3% in 2009 to 28.7% in 2011), and although emaciation and weight deficit were not significantly reduced, there was a slight improvement (13.5% to 12.4 and 23.7% to 21.1%, respectively) [64].

In addition to the pedagogical tools of Positive Deviance/Hearth, nutritional supplementation has a considerable impact on improving anthropometric development [65,66,67].

Of all the studies on nutritional supplementation with micronutrients and fortification of local foods, just one has reported an effect on anthropometric changes and child malnutrition [38]: Seetha et al. 2018 [37], who gave theoretical-practical workshops on complementary feeding and hygiene and provided a mix of flours in a quasi-experimental clinical trial. Indicators such as arm circumference, W/H and W/A underwent positive modifications. This study used the Positive Deviance/Hearth approach in addition to supplementation and showed a considerable improvement in anthropometric development.

Of the programs that encouraged domestic agriculture, Gelli et al. (2018) [39] obtained anthropometric effects in H/A (intervention −1.87 vs. control −2.29; *p* ≤ 0.05) and a reduction in chronic malnutrition (intervention 41 to 45% vs. control 42 to 63%; *p* ≤ 0.05) by focusing on improving the techniques of food production, animal care and seed and animal delivery for farmers, parents and teachers in nursery schools. Although other studies that promoted agriculture such as those by Kuchenbecker et al. (2017) [32] and Olney et al. (2015) [20] did not generate statistically significant anthropometric effects, the H/A and chronic malnutrition indicators presented slightly favourable results in the intervention groups.

Overall, we observe that nutrition education programs aimed at mothers or caregivers of children show a positive effect both on the mother’s knowledge and eating practices, as well as on nutritional intake and child growth and development. However, although the population group to which it is directed responds easily to interventions, nutrition education by itself does not change underlying conditions such as poverty and lack of food and healthcare resources that contribute to poor child growth [9].

## 5. Strengths and Limitations

The bibliographic search was carried out from the latest revision available in the literature [68].

The scientific evidence found would have been greater if we had selected publications from a longer period. However, this research is expected to stimulate the implementation of health strategies based on interventions with impact evaluation, promote the development of cost-effective health interventions and optimize the use of resources in nutrition and maternal and child health.

There is possibly a high heterogeneity in the articles included, since the intervention programs have different approaches (educational programs, educational and supplementation programs and agricultural and educational programs) and different evaluations of their effect.

## 6. Policy and Future Research

A recent systematic review [69] assesses the impacts (health, social, economic and environmental) of community food production initiatives in several African countries (Kenya, Cameroon and South Africa). Most of the studies reviewed focused on economic and environmental impacts, with very limited evidence of the impact of food production on food security, nutritional status and dietary intake. The authors highlighted the need for research to base its design on explicit theoretical frameworks, including various approaches, so that the interpretation of the results could be more useful for carrying out coordinated intersectoral actions and policy initiatives aimed at improving food security and nutrition.

In this regard, FAO’s African Regional Overview on Food Security and Nutrition [FAO, ECA, AUC] also indicates the need for coordinated policy initiatives across the entire food system, from food production to marketing and consumer demand, to identify more effective systems [70].

The Foresight Africa project consists of a series of reports on the main points to take into account in the development of effective policies to reduce malnutrition on this continent, explaining the points they consider to be priorities. These reports highlight the importance of coordinating scientific evidence with the activities of farmers to generate more effective solutions, adapted to the demand of the community. They also emphasize the emphasis on the application of digital technologies in food production, which allow preventing climate crises, improving irrigation systems, etc., as well as improving the use and safety of food, among others. These reports also consider that it is not possible to address food insecurity without addressing the causes of conflict and fragility related to agriculture and recommend strengthening the capacity of local institutions to promote the design and implementation of community-based approaches, making better use of natural resources in these fragile areas [3].

## 7. Conclusions

Although the studies on nutrition education to combat child malnutrition in Africa found common ground, joint evaluation of the results of the different programs is difficult given their methodological diversity and outcome variables. However, it should be noted that although the interventions on nutrition education carried out on mothers and caregivers were useful for improving dietary diversity and attitudes toward recommended feeding, they appear to be insufficient to generate nutritional changes in children. For this reason, the programs should be accompanied by actions of national agriculture or nutritional supplementation. Only then will people be able to adhere to the recommendations and guidelines of a healthy diet to stop the processes of child malnutrition since food shortages are a latent problem in Africa. Despite the observed benefits, more studies are required to provide more evidence on the characteristics of the most effective interventions.

## Figures and Tables

**Figure 1 ijerph-18-07709-f001:**
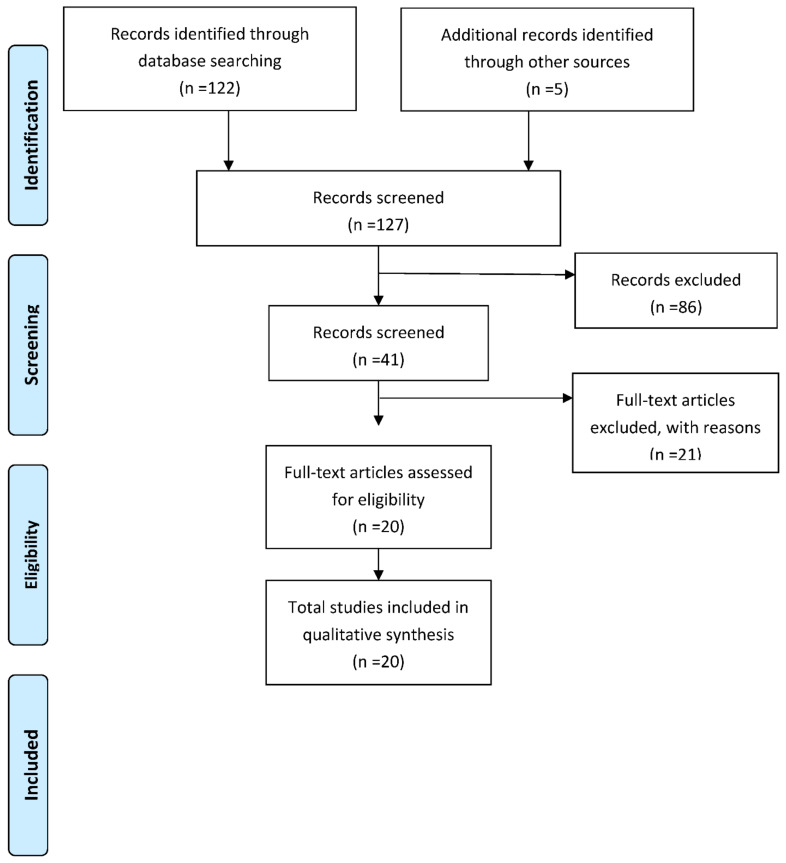
Flow diagram.

**Table 1 ijerph-18-07709-t001:** Main results of nutrition education programs for African mothers.

Author Ref. Year, Country	Type of Intervention	Intervention Methodology	Nutrition Indicators	Results	Q
Negash et al. 2014. Ethiopia [11]	RCT, EP Educational model: Workshops on infant feeding, practical demonstrations of supplemented baby food recipes. Visual material Alive and Thrive [12]	Realized by nutrition educators. IG N = 80/CG N = 73 Directed to mothers of infants 6 to 23 months of age (N = 153) Format: Group 12 sessions (twice a month; 120 min)	At 6 months post-intervention Boy: MDD and MMF Anthropometry, food consumption Mother: nutritional knowledge. Complementary feeding practices	IG N = 80/CG N = 73 Knowledge: 5.8 to 7.11 **/6.3 to 6.3 Complementary Feeding: 9.1 to 9.6 */9.1 to 9.1 Minimum Diet Diversity: 33 to 54%/32 to 14% Minimum meal frequency: 28 to 52%/25 to 32% Energy: 854 to 1045 Kcal/717 to 885 Kcal Protein: 24 to 28.7 g/20.6 to 21.6 *p* ≤ 0.05 Iron: 28.9 to 30.6 mg/16.9 to 20.9 *p* ≤ 0.05 Non-significant changes in nutritional classification	Q4
Desai et al. 2015. Zimbabwe [13]	Quasi-experimental, EP Educational model: home visits on maternal and infant feeding and nutrition, delivery tools and activities to illustrate key concepts, fortified recipes.	Realized by: Health workers Directed to mothers of infants 6 to 12 months of age (*N* = 19) Format: individual home visits 4 month	Infants: food consumption Mother: nutritional knowledge	PRE-TEST-FINAL TEST (%met requirement) Energy 1759 (63%)–2635 (79%) *p* < 0.05 Protein 9.9 g (84%)–15.8 g (95%) *p* < 0.05 Fat 9.3 g (89%)–20 g (100%) *p* < 0.05 Vit. A 58 µg (79%)–678 µg (100%) *p* < 0.05 Folate 26.4 µg (31%)–119 µg (68%) *p* < 0.05 Calcium 58 mg (10%)–352 mg (89%) *p* < 0.05 Iron 2.4 mg (0%)–8.9 (68%) *p* < 0.05 Zinc 2.3 mg (16%)–11 (89%) *p* < 0.05	Q1
Waswa et al. 2015. Kenya [14]	RCT, EP Educational model: Workshops in infant feeding, practical demonstrations of cooking and recipes, practice food hygiene. FAO and UNICEF [15,16]	Realized by: Health workers Directed to mothers of infants 6 to 17 months of age (N = 207). Format: group and individual 4 sessions 2–5 h	At 12 months post-intervention. Infants: Anthropometry. MDD and MMF. Mother: nutritional knowledge	IG N = 110/CG N = 97 H/A −1.61 to −1.85/−1.31 to −1.50 W/A −0.87 to −0.74/−0.57 to −0.39 *p* = 0.022 W/H −0.04 to 0.25/0.15 to 0.49 Chronic malnutrition 29.3 to 49%/29.3 to 34% Low weight 17.2 to 8.2%/10.1 to 7.2% Acute malnutrition 2 to 0%/2 to 3.1% Minimum dietary diversity: 55.6 to 87.3%/50.5 to 55.7% *p* ≤ 0.001 Minimum meal frequency: 80.6 to 98.8%/88.9 to 88.6% *p* = 0.01 Nutritional knowledge: 8.21/21/3.66/21 *p* ≤ 0.001	Q3
Tariku et al. 2015. Ethiopia [17]	RCT, EP Educational model: Workshops on infant food and nutrition health believe model [18,19]	Realized by: Health workers and community volunteers Directed to Women with infants 6 to 18 months of age (N = 166) Format: 2 sessions/month for 3 months Group and individual	At the end of the intervention Infants: MDD and MMF	HBM N = 56^A^/TRADITIONAL METHOD N = 54^B^/CG N = 56^C^ Minimum dietary diversity: versus AB and AC. *p* ≤ 0.001 Minimum meal frequency: versus AB and AC *. *p* ≤ 0.005	Q4
Olney et al. 2015. Burkina Faso [20]	RCT, AEP Contents: Domestic agriculture. Workshops on maternal and infant feeding and nutrition. Helen Keller International [21]	Carried out by: Leader mothers (LM +), health committee (HC) Directed to mothers of infants 3–12.9 months of age (N= 1481) Format: individual. Home visits 2 times/month for 6 months.	At 2 years post-intervention Infants: MDD, anthropometry and haemoglobin Mother: Knowledge	LM + N= 443^A^/HC = 441^B^/GC N = 597^C^ H/A −1.06 to −1.77/−1.35 to −1.96/−1.29 to −1.91 Chronic malnutrition 28.5 to 43.5%/33.5 to 47%/30.3 to 47.7% W/H −0.98 to −0.66/−1.16 to −0.73/−0.96 to −0.66 Acute malnutrition 25.7 to 8.4%/30.8 to 8.6%/24.3 to 10.2% W/A −1.41 to −1.44/−1.73 to −1.62/−1.63 to −1.53 Under weight 31.9 to 26.5%/41.5 to 31.3%/39.5 to 31.4% Knowledge 34 to 75%/38 to 79%/42 to 63% *p* ≤ 0.05 ABC MDD 3% to 15%/1.7% to 18.2%/2.6% to 6.3% Haemoglobin 9 to 9.4 g/dl/8.87 to 9.87 g/dl/9.3 to 9.5 g/dl *p* ≤ 0.05^CB^ Adjusted at 3–5.9 months of age	Q1
Mulualem et al. 2016. Ethiopia [22]	RCT, EP Educational model: Workshops on infant food and nutrition. Health believe model and Alive-Thrive manual [18,23]	Realized by: Health workers Directed to mothers of infants 6 to 18 months of age (N = 160) Format: group and individual 2 sessions/month for 6 months	At 2 weeks post-intervention Infants: Anthropometry MMF Mother: Complementary feeding practices, nutritional knowledge	IG N = 80/CG N = 80 Nutritional knowledge 1.09 to 9.46 */1.48 to 1.68 *p* ≤ 0.05 Complementary feeding 1.31 to 7.6 */1.15 to1.23 *p* ≤ 0.05 W/H 0.42 to 2.20 */0.37 to 0.89 *. *p* ≤ 0.05 H/A −1.34 to −2.78 */−1.43 to −2.82 * W/A −0.39 to 0.31 */−0.43 to −0.73 * *p* ≤ 0.05 MUAC 13.09 to 13.34/13.06 to 13.19 Minimum meal frequency 86.2 to 83.3%/81.2 to 52.5%	Q4
For the PROMISE-EBF Study Group et al., 2016. Uganda [24]	RCT, EP Educational model: Support workshops on breastfeeding, infant feeding and HIV. WHO [25,26]	Directed to Pregnant women and mothers of infants <6 months of age (Base N = 886; final line N = 466) Format: Individual 5 home visits/month during pregnancy and 4 visits after birth.	5 years post-birth age Infants: anthropometry.	IG/CG H/A −0.45 to −1.78/−0.32 to −1.53 W/A −0.40 to −1.28/−0.16 to −1.06 Chronic malnutrition 12 to 41%/7 to 33% Low weight 8 to 26%/5 to 16%	Q2
Kang et al. 2017. Ethiopia [27]	RCT, ESP Educational model: Workshops on infant feeding, practical demonstrations of recipes. Positive Deviance (PD)/Hearth [28,29,30]	Realized by community volunteers or health workers. Directed to Mothers of children 6 to 24 months of age (N = 1790) Format: group and individual 12 sessions over 12 months.	12 months after the intervention. Infants: Anthropometry	IG N = 876/CG N = 914 Monthly growth effect H/A −0.074/−0.095. *p* = 0.001 W/A −0.032/−0.060. *p* ≤ 0.001 W/H 0.011/0.030. *p* ≤ 0.001	Q3
Kang et al. 2017. Ethiopia [31]	RCT, ESP Educational model: Workshops in infant feeding, practical demonstrations of recipes. Positive Deviance/Hearth [28,29,30]	Realized by community volunteers or health workers. Directed to Mothers of infants 6 to 24 months of age (N = 1199) Format: group and individual 12 sessions over 12 months.	12 months after the intervention. Infants: MDD and MMF Mother: Handwashing.	IG N= 570/CG N= 629 Minimum meal frequency: 7.95 of 15/6.8 of 15 *p* = 0.003 Minimum dietary diversity: 4.68 out of 10/4.40 out of 10 Handwashing: 3.07 out of 6/2.64 out of 6	Q3
Kuchenbecker et al. 2017. Malawi [32]	RCT, AEP Educational model: Workshops on infant feeding, agricultural and livestock supplies, food hygiene practices, practical cooking demonstrations and recipes. FAO and UNICEF [33,34]	Realized by community volunteers Directed to agricultural mothers of infants 6 to 23 months of age (Base N = 832; final line N = 959) Format: Group 10 sessions of 2–3 h during 5 months	At 36 months post-intervention Infants: Anthropometry. MDD and MMF	IG/CG W/A −0.93 to −0.69/−0.86 to −0.76 H/A −1.81 to −1.79/−1.71 to −1.85 W/H 0.01 to 0.32/0.03 to 0.27 Minimum dietary diversity 61.9 to 71.1%/59.9 to 55.5% *p* = 0.01 Minimum meal frequency: 88.6 to 90.3%/80.4 to 81.6%	Q2
Muhoozi et al. 2018 Uganda [35]	RCT, EP Educational model: food hygiene practices, infant feeding workshops, practical cooking and recipe demonstrations. WHO [36]	Realized by leaders in health team Directed to mothers of infants 6 to 8 months of age (N = 511) Format: group and individual 3 sessions (6–8 h/session)	Approx. 20–24 months old participants. Infants: Anthropometry	IG N = 243/CG N = 224 H/A −1.07 to −2.15/−1.2 to −2.25 W/A −0.63 to −0.87/−0.72 to −0.88 W/H 0.12 to 0.31/0.16 to 0.36 HCZ: 0.68 to 0.39/0.57 to 0.33	Q2
Seetha et al. 2018. Malawi [37]	Quasi-experimental, ESP Educational model: Workshops on infant feeding, practice food hygiene, provision of flour mix, practical cooking demonstrations and recipes. Positive Deviance/Hearth [38]	Realized by: Health workers Directed to mothers of infants <24 months of age (N = 179). Format: group 21 sessions, 3 h	At the end of the intervention Infants: Anthropometry.	RANDOM EFFECT W/H −0.70 to 0.85 *p* = 0.002 W/A −0.25 to 0.73 *p* = 0.000 H/A 0.59 to 0.18 *p* = 0.458	Q3
Gelli et al. 2018b. Malawi [39]	RCT, AEP Educational model: Workshops in infant feeding, practical demonstrations of cooking and recipes, food hygiene practices, domestic agriculture [40,41,42]	Realized by: Government trainers. Directed to farmers, parents, teachers in nursery schools. Format: group and individual 2 weeks training + 3 days/month. Monthly follow-ups.	At 12 months post-intervention Infants: Anthropometry Infants: Food consumption	Infants 6–24 months of age IG N = 155/CG N = 149 H/A −1.70 to −1.87/−1.61 to −2.29 *p* ≤ 0.05 Chronic malnutrition 41 to 45%/41 to 63%. *p* ≤ 0.05 W/H 0.12 to 0.04/0.09 to 0.09 Acute malnutrition 1 to 2%/3 to 1% W/A −0.68 to −1.05/−0.73 to −1.18 Low weight 14 to 16%/13 to 22% Participant 36–72 months of age N = 631/N = 617 H/A −1.75 to −1.70/−1.74 to −1.70 Chronic malnutrition 40 to 36%/39 to 36% W/H 0.09 to −0.06/0.11 to 0.08 Acute malnutrition 1 to 1%/2 to 1% W/A −1.08 to −1.16/−1.05 to −1.15 Low weight 17 to 34%/17 to 32% Energy 1273 to 1627 kcal/1321 to 1376 *p* ≤ 0.001 Protein 40 to 54 g/42 to 48 g *p* ≤ 0.05 Vitamin A 449 to 930 µg/600 to 1000 µg Iron 11 to 13 mg/11 to 12 mg. *p* ≤ 0.05	Q1
Somassè et al. 2018. Mali [43]	RCT, ESP Contents: Micronutrient supplementation and workshops on infant food and nutrition. WHO [36,44]	Realized by: community volunteers Directed to mothers of children 6–23 months of age (N = 722) Format: micronutrient supplementation and nutritional education (2 sessions/month) for 3 months	3 months after the intervention Child: Anthropometry, haemoglobin	IG N = 396/CG N = 326 Weight change 0.76 kg/0.74 kg MUAC 3.4 mm/3.8 mm Change length 3 cm/2 cm Hemoglobin change 0.50 g/dl/0.9 g/dl *p* = 0.023	Q3
Mbogori et al. 2019, Kenya [45]	Quasi-experimental, EP Educational model: Workshops on infant food and nutrition, practical cooking demonstrations, food hygiene practices and child care [46]	Realized by women leaders and researchers Directed to mothers of infants <5 years (Mothers N= 48; Infants N = 45) Format: Group 5 days (two sessions/day 120–180 min)	At 6 months post-intervention. Infants: food consumption Infants: anthropometry, nutrition knowledge	Pre-test–post-test Vitamin A: 28.6–116.8 µg retinol *p* = 0.001 Calcium: 74.6–173.6 mg *p* = 0.001 Knowledge: 68–91% *p* = 0.004 Energy: 755–636 kcal Protein: 19.7–14.7 g Acute malnutrition 21.7–26% Chronic malnutrition 29–19% Low weight 29–22% Non-significant changes: W/H, H/A and W/A	NIJ
Kajjura et al. 2019, Uganda [47]	RCT, EP Educational model: practical food hygiene and workshops on complementary feeding and infant nutrition, practical cooking and recipe demonstrations. Health believe model [48,49]	Realized by Health workers aimed at mothers of infants 6 to 18 months of age, moderate acute malnutrition (N = 204) Format: Group 12 sessions (weekly for 3 months, 60 min)	At the end of the intervention Infants: MDD and MMF practices. Mother: knowledge of MDD, MMF, food hygiene.	IG N = 104 (supplemented porridge: sorghum-based malted porridge)/CG N = 100 (porridge supplemented with: soy and corn mix) Practices: Minimum meal frequency: 41 to 83.7%/40 to 93%. *p* = 0.038 Minimum dietary diversity: 8.7 to 77.9% **/18 to 88% ** N = 204 Base/post-intervention: Practices: Minimum meal frequency: 40.7% to 88.2% *p* ≤ 0.001 Minimum dietary diversity: 13.2% to 82.8% *p* ≤ 0.001 Food hygiene: 36.7% to 90.2% *p* ≤ 0.001 Knowledge: Minimum meal frequency: 2.21 to 2.82 **/2.1 to 2.83 ** Minimum dietary diversity: 3.76 to 5.63 **/3.76 to 6.14 ** Food hygiene: 2.13 to 3.31 **/2.08 to 3.52 **	Q4
Atukunda et al. 2019a. Uganda [50]	RCT, EP Educational model: food hygiene practices, infant feeding workshops, practical cooking and recipe demonstrations. WHO [36]	Realized by leaders in health team Directed to mothers of infants 6 to 8 months of age (N = 511) Format: group and individual 3 sessions (6–8 h/session)	Approx. 20–24 months old participants. Infants: Anthropometry	Baseline IG N = 263/CG N = 248 at 20–24 months of age IG N = 77/CG N = 78 Chronic malnutrition 20.9 to 41.6%/28 to 59% Low weight 9.5 to 7.8%/14.5 to 10.3% Acute malnutrition 4.6 to 3.9%/4.8 to 2.6%	Q1
Atukunda et al. 2019b. Uganda [51]	RCT, EP Educational model: food hygiene practices, infant feeding workshops, practical cooking and recipe demonstrations. WHO [36]	Realized by leaders in health team Directed to mothers of infants 6 to 8 months of age (N = ±150) Format: group and individual 3 sessions (6–8 h/session)	Approx. 20–24 to 36 months of age Infants: Anthropometry.	Baseline IG N = 263/CG N = 248 at 20–24 months age IG N = 74–77/CG N = 73–78 Chronic malnutrition 20.9 to 18.1%/28 to 36% Low weight 9.5 to 8.3%/14.5 to 11.3% Acute malnutrition 4.6 to 4.2%/4.8 to 2.8% Growth from 20–24 months to 36 months old IG N= 74–77/CG N= 73–78 H/A −1.96 to −2.15/−2.07 to −2.65 *p* = 0.0001 W/A −0.76 to −0.98/−0.85 to −1.18 *p* = 0.40 W/H 0.26 to 0.44/0.45 to 0.84 *p* = −0.054 HCZ 0.30 to −0.34/0.61 to 0.05 *p* = 0.055	Q1
Hitachi et al. 2020. Kenya [52]	Quasi-experimental, EP Educational model: educational sessions maternal and child nutrition and follow-up consultations. WHO [53]	Realized by: community health workers Directed to household and children aged 6–59 months Format: educational sessions on maternal nutrition during pregnancy and lactation, breastfeeding, complementary feeding, diverse diet and food groups, hygiene and sanitation practices, supplements health program and family planning.	At the end of the intervention Household: diet quality, food consumption Child: Anthropometry	CG: N = 181 household; N = 113 children/ IG: N = 181 household; N = 67 children Household DDS 4.18 to 6.15 *p* ≤ 0.01/4.97 to 6.91 *p* ≤ 0.01 Child anthropometry H/A −1.56 to −1.55/−1.30 to −1.51 W/A −1.14 to −1.04/−0.89 to −0.97 W/A −0.44 to −0.22 *p* = 0.06/−0.24 to −0.16	Q2
Leroy et al. 2020. Burundi [54]	RCT, ESP Contents: 3 treatment arms and 1 control arm: 1 treatment arm (from pregnancy to 18 months); 2 treatment arm (from pregnancy to 24 months); 3 treatment arm (from birth to 24 months). Food ration, health services and behaviour change communication.	Directed to child 0–24 months Format: 3 treatment arms received household and individual (mother or child in the first 1000 days) food rations (corn-soy blend and micronutrient-fortified vegetable oil), The control arm received no rations or behaviour change communication.	At the end of the intervention Child: Anthropometry.	1 treatment arm: N = 866; 2 treatment arm: N = 425; 3 treatment arm: N = 420/CG: N = 855 W/L 1 treatment arm: −0.3 to −0.1 W/L 2 treatment arm: −0.3 to −0.2 W/L 3 treatment arm: −0.3 to −0.3 W/L CG: −0.3 to −0.3	Q1

Abbreviation: NIJ: non-indexed journal. RCT: randomized clinical trial. EP: educational programs. ESP: educational and supplementation programs. AEP: agricultural and educational programs. IG: intervention group. CG: control group. HCZ: head circumference. W/H: weight-for-height. H/A: size-for-age. W/A: weight-for-age. MUAC: arm circumference. HBM: health believe Model. MDD: minimum dietary diversity > 4 meals/day. MMF: minimum meal frequency is 3–4/day. DDS: dietary diversity score. *p* values with significant intergroup difference (control group vs. intervention group). *, **: values with significant intra-group difference *: *p* < 0.05; **: *p* < 0.01. Chronic malnutrition (<−2 DS Z-SCORE H/A). Low weight for age (<−2 DS Z-SCORE W/A). Acute malnutrition (<−2 DS Z-SCORE W/H or W/L). Q: impact index of the journal.

## Data Availability

Data sharing is not applicable to this article as no new data were created or analyzed in this study.

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
