# Peer review of "Nutrition Education Programs Aimed at African Mothers of Infant Children: A Systematic Review"

_ijerph, 2021, doi:10.3390/ijerph18147709_

Round 1
Reviewer 1 Report
The current manuscript entitled ‘Nutrition Education Programs aimed at African mothers of infant children: A systematic review’ is a well-written review article. I have few minor concerns to improve the quality of the manuscript. Since authors have conducted their systematic studies till 2020, In the introductory section, authors might consider adding the latest statistic on malnutrition. In the materials and methods section, please provide a tabular format for the intervention programs chosen. In the results section, change the word ‘returned’ to retrieved.
Author Response
Answers to Reviewer 1: Dear Reviewer 1, Thank you for reviewing our manuscript (ijerph-1237368) titled: "Nutrition Education Programs aimed at African mothers of infant children: A systematic review".We found your contributions very interesting and useful, so the text has been modified in relation to your suggestions. We highlighted in boldface the changes made in the text. Here we detail the changes made with the answers corresponding to your comments point by point:
- Reviewer 1: The current manuscript entitled ‘Nutrition Education Programs aimed at African mothers of infant children: A systematic review’ is a well-written review article. I have few minor concerns to improve the quality of the manuscript. Since authors have conducted their systematic studies till 2020, In the introductory section, authors might consider adding the latest statistic on malnutrition
Answer: We agree with his point of view. We have revised introductory section and we have considered adding the latest stadistic on malnutrition (Page 2, Lines 40-45).
- Reviewer 1: In the materials and methods section, please provide a tabular format for the intervention programs chosen.
Answer: We agree with his point of view. We have revised materials and methods section and we have improved selection criteria (Page 3, Lines 82-96).
- Reviewer 1: In the results section, change the word ‘returned’ to retrieved.
Answer: We agree with his point of view. We have changed the word “returned” to retrieved (Page 4, Line 113).

Reviewer 2 Report
Thank you for giving me the opportunity to review this article. This review aims to elucidate the effect of intervention programs in nutrition education, designed for African mothers, on the nutritional status of children of infant age. It is a very interesting review, nevertheless I have some doubts and recommendations
-I strongly recommend following the PRISMA guideline for systematic reviews.
Introduction
- What the authors consider as infant age?
Methods
- If the authors tried to elucidate the effects of nutrition education programs, why do they only consider articles from 2015 to 2020? I strongly recommend extending the time range, as the authors conclude “the evidence would have been greater if we had selected publications from a longer period”.
- Please clarify the inclusion and exclusion criteria.
- Please clarify the filters and limits used in the search strategy.
- Specify the methods used to assess risk of bias in the included studies
- Please clarify the processes used to decide which study is eligible for each synthesis.
- Please specify the date when the source was last consulted.
Results
- Present assessments of risk of bias for each included study.
- Present results of all investigations of possible causes of heterogeneity among study results
- Please name abbreviations before using them.
Discussion
- Discuss any limitation of the evidence included in the review. The heterogeneity of the results is a limitation to extract any type of conclusion.
- Discuss implications of the results for practice, policy and future research.
- Line 205 please change “the results obtained here”.
- Line 231, how long was the intervention?
Author Response
Answers to Reviewer 2: Dear Reviewer 2, Thank you for reviewing our manuscript (ijerph-1237368) titled: "Nutrition Education Programs aimed at African mothers of infant children: A systematic review".We found your contributions very interesting and useful, so the text has been modified in relation to your suggestions. We highlighted in boldface the changes made in the text. Here we detail the changes made with the answers corresponding to your comments point by point: 1. Reviewer 2: I strongly recommend following the PRISMA guideline for systematic reviews.Answer: We agree with his point of view. We have adapted the article according to the PRISMA guideline 2020.Reference: Page MJ, McKenzie JE, Bossuyt PM, Boutron I, Hoffmann TC, Mulrow CD, et al. The PRISMA 2020 statement: an updated guideline for reporting systematic reviews. BMJ 2021;372:n71. doi: 10.1136/bmj.n71.
- 2. Reviewer 2: Introduction; What the authors consider as infant age?
Answer: The authors consider infant age as those under 5 years of age.
- Reviewer 2: Methods; If the authors tried to elucidate the effects of nutrition education programs, why do they only consider articles from 2015 to 2020? I strongly recommend extending the time range, as the authors conclude “the evidence would have been greater if we had selected publications from a longer period”.
Answer: In accordance with its indications, we have increased the search for manuscripts until October 2012, when a systematic review of intervention studies was carried out through randomized clinical trials (Lazzerini M, Rubert L, Pani P. Specially formulated foods for treating children with moderate acute malnutrition in low‐ and middle‐income countries. Cochrane Database of Systematic Reviews 2013, Issue 6. Art. No.: CD009584. DOI: 10.1002/14651858.CD009584.pub2), to assess the safety and effectiveness of different types of infant feeding in children with moderate acute malnutrition in low- and middle-income countries. This review included 11 manuscripts of which 7 were made in African countries.
The researchers indicate that the interventions in this field had not been aimed at improving the family diet and recommended that the following intervention programs be aimed at promoting food and nutrition education, as is the case in our review. Therefore, it is not necessary to extend the search to dates prior to the review by Lazzerini et al. (2013)
According to the changes made, the new data has been introduced in the text: abstract, figure 1, materials and methods, results and discussion.
- Reviewer 2: Methods; Please clarify the inclusion and exclusion criteria
Answer: We agree with his point of view. We have revised materials and methods section and we have improved selection criteria (Page 3, Lines 82-86).
- Reviewer 2: Methods; Please clarify the filters and limits used in the search strategy
Answer: We agree with his point of view. We have revised materials and methods section and we have clarified the filters and limits used in the search strategy (Page 3, Lines 94-96).
- Reviewer 2: Methods; Specify the methods used to assess risk of bias in the included studies.
Answer: Although the bias risk assessment is interesting, we have not assessed the potential for publication bias using funnel plots and the tests of Begg and Egger since it is a systematic review and not a meta-analysis.
References:
- Begg CB, Mazumdar M. Operating characteristics of a rank correlation test for publication bias. Biometrics 1994; 50: 1088–101. 33.
- Egger M, Davey Smith G, Schneider M, Minder C. Bias in meta–analysis detected by a simple, graphical test. BMJ 1997; 315: 629–34.
- Reviewer 2: Methods; Please clarify the processes used to decide which study is eligible for each synthesis.
Answer: We agree with his point of view. We have revised materials and methods section and we have clarified the processes used to decide wich study was eligible for each synthesis (Pages 3-4, Lines 100-107).
- Reviewer 2: Methods; Please specify the date when the source was last consulted.
Answer: We agree with his point of view. We have added in the search strategy when was the last consulted (Page 3, Line 75).
- Reviewer 2: Results; Present assessments of risk of bias for each included study.
Answer: Although the bias risk assessment is interesting, we have not assessed the potential for publication bias using funnel plots and the tests of Begg and Egger since it is a systematic review and not a meta-analysis.
References:
- Begg CB, Mazumdar M. Operating characteristics of a rank correlation test for publication bias. Biometrics 1994; 50: 1088–101. 33.
- Egger M, Davey Smith G, Schneider M, Minder C. Bias in meta–analysis detected by a simple, graphical test. BMJ 1997; 315: 629–34.
- Reviewer 2: Results; Present results of all investigations of possible causes of heterogeneity among study results.
Answer: We consider that there is a high heterogeneity in the articles included, since there are different approaches in the intervention programs, some only educational, others educational and agriculture and supplementation programs. On the other hand, their evaluation focuses on different nutritional aspects of mothers or caregivers (nutritional education) and children (food consumption, anthropometric measurements, malnutrition rates). However, since it is not a meta-analysis, we have not quantitatively calculated heterogeneity.
- Reviewer 2: Results; Please name abbreviations before using them.
Answer: We have added the name abbreviations in the results section (Page 6, Lines 167-168 and 186).
- Reviewer 2: Discussion; Discuss any limitation of the evidence included in the review. The heterogeneity of the results is a limitation to extract any type of conclusion.
Answer: As we have already indicated, we believe that there is high heterogeneity in the manuscripts included, which we indicate in the limitations (Page 12, Lines 367-369).
- Reviewer 2: Discussion; Discuss implications of the results for practice, policy and future research.
Answer: We agree with his point of view. We have added a section with implications of the results for practice, policy and future research. (Page 9, Line 227).
- Reviewer 2: Discussion; Line 205 please change “the results obtained here”.
Answer: We agree with his point of view. We have changed in the discussion section (Page 9, Line 227).
- Reviewer 2: Discussion; Line 231, how long was the intervention?
Answer: We agree with his point of view. We have changed in the discussion section (Page 9-10, Lines 251-252).

Reviewer 3 Report
The methodology section is clearly defined. Figure 1 supports how each article was selected for the review
Results are difficult to generalize given the broad differences in the articles, but the authors made attempts to fully describe the similarities and differences between articles.
I appreciate that authors recognize the limitation of only reviewing a period of 5 years of research. The results may have been strengthened by reviewing a larger time frame.
The article needs a substantial review and editing. For example on lines 351-352, “...although nutrition education interventions for mothers and caregivers are useful tools for nutrition education”
Author Response
Answers to Reviewer 3: Dear Reviewer 3, Thank you for reviewing our manuscript (ijerph-1237368) titled: "Nutrition Education Programs aimed at African mothers of infant children: A systematic review".We found your contributions very interesting and useful, so the text has been modified in relation to your suggestions. We highlighted in boldface the changes made in the text. Here we detail the changes made with the answers corresponding to your comments point by point:
- Reviewer 3: The methodology section is clearly defined. Figure 1 supports how each article was selected for the review. Results are difficult to generalize given the broad differences in the articles, but the authors made attempts to fully describe the similarities and differences between articles. I appreciate that authors recognize the limitation of only reviewing a period of 5 years of research. The results may have been strengthened by reviewing a larger time frame.
Answer: In accordance with its indications, we have increased the search for manuscripts until October 2012, when a systematic review of intervention studies was carried out through randomized clinical trials (Lazzerini M, Rubert L, Pani P. Specially formulated foods for treating children with moderate acute malnutrition in low‐ and middle‐income countries. Cochrane Database of Systematic Reviews 2013, Issue 6. Art. No.: CD009584. DOI: 10.1002/14651858.CD009584.pub2), to assess the safety and effectiveness of different types of infant feeding in children with moderate acute malnutrition in low- and middle-income countries. This review included 11 manuscripts of which 7 were made in African countries.
According to the changes made, the new data has been introduced in the text: abstract, figure 1, materials and methods, results and discussion.
- Reviewer 3: The article needs a substantial review and editing. For example on lines 351-352, “...although nutrition education interventions for mothers and caregivers are useful tools for nutrition education”
Answer: We agree with his point of view. The article has been reviewed by an english reviewer to improve understanding and quality.
We have changed in the conclusion section (Page 11, Lines 377-379).

Round 2
Reviewer 2 Report
Thank you for giving me the opportunity to review this article. This review aims to elucidate the effect of intervention programs in nutrition education, designed for African mothers, on the nutritional status of children of infant age. It is a very interesting and useful review. The authors have appropriately modified the review according the reviewers suggestions.
Reviewer 3 Report
I believe the edits have strengthened this article. I think it is more clear.